# Prevalence of early childhood caries, risk factors and nutritional status among 3-5-year-old preschool children in Kisarawe, Tanzania

**Tumaini S. Ndekero**[1☯], **Lorna C. Carneiro**[1☯]*, **Ray M. Masumo**[2☯]

**1** Department of Restorative Dentistry, Muhimbili University of Health and Allied Sciences, Dar es salaam, Tanzania, **2** Oral Health Section, Ministry of Health, Community Development, Gender, Elderly and Children, Dodoma, Tanzania

☯ These authors contributed equally to this work.

* carneiro2@hotmail.com

## Abstract

**Data Availability Statement:** All relevant data are within the manuscript and its Supporting information files.

### Background

Early childhood dental caries (ECC), a serious public health problem lacks information on the association between ECC, risk factors and nutritional status among children in Tanzania. Therefore, this study aimed to determine the prevalence of ECC, risk factors and nutritional status among 3-5-year-old preschool children and to assess its correlation with the form, visible plaque scores in upper anterior teeth, total sugar exposure, anthropometric measures, and socio demographic attributes.

### Methods

This cross-sectional study was conducted on 831 children registered in public preschools in Kisarawe District. Assessment of ECC used the WHO (2013) criteria and anthropometric measures used the WHO Child Growth Standards (2006). Structured questionnaires were completed by children's parents through an interview. Collected information on socio-demographic attributes including oral hygiene and sugar exposure to their children was analyzed.

### Results

Only 459 children (55.2%) were recognized as caries-free. Dental caries experience in terms of decayed, missing and filled teeth (dmft) index was 2.51. Children with visible plaque were the majority (56.1%). The dmft score index was negatively and significantly associated with weight-for-age z-score [Coefficient: -0.11 (95% CI: -0.76, -0.11)] and positively significantly associated with visible plaque score index [Coefficient: 0.16 (95% CI: 0.18, 0.52)] and total sugar exposure [Coefficient: 0.19 (95% CI: 0.15, 0.38)] in the adjusted model. The prevalence of underweight was 4.2% [95% CI: (3.0–5.8)] and severe underweight was 0.2% [95% CI: (0.1–0.9)] while prevalence of stunting was 1.6% [95% CI: (0.9–2.7)] and severe stunting was 0.4% [95% CI: (0.1–1.1)].

**Funding:** Financial support with the aim to uplift junior staff (TSN) in data collection was received from the Muhimbili University of Health and Allied Sciences (institution) to support the purchase of materials, supplies and equipment for data collection including transport expenses to study venue. Authors, (TSN and LCC) are employed by the institution. The funders had no role in study design, data collection and analysis, decision to publish, or preparation of the manuscript.

**Competing interests:** The authors have declared that no competing interests exist.

## Conclusion

This study demonstrated a significant negative relationship between ECC and children's anthropometric measures indicated by weight-for-age, and positive relationship with sugar exposure and poor oral hygiene indicated by visible plaque on upper anterior teeth. Controlling risk factors will reduce the occurrence of ECC thereby catering for a healthy population of well-nourished children.

## Introduction

Early Childhood Caries (ECC) to 3–5 years old is a major public health concern not only in most European countries and the USA but also in disadvantaged communities in both developing and industrialized countries [1,2]. A systematic analysis for the global burden of diseases, injuries and risk factors for 195 countries from 1990 to 2015 reported that almost 8% of children globally were affected by untreated ECC [3].

While some studies have evaluated and categorized the risk factors of ECC to be related to sociodemographic factors, dietary factors and oral hygiene factors [4–6] other studies report familial socioeconomic background, lack of parental education, and lack of access to dental care to also be contributing factors for the high prevalence of ECC [7]. However, the degree to which different risk factors are associated with ECC remains unclear [1–3].

ECC has been associated with other health problems, ranging from local pain, infections, abscesses, leading to difficulty in chewing, malnutrition and difficulty in sleeping [8]. Furthermore, the associated pain from dental caries has a negative impact on children's emotional status, sleep patterns, and ability to learn or perform their usual activities [2]. Preschool children with ECC, if left untreated, might experience dental pain leading to avoidance of certain types of foods which might interfere adversely with their nutritional status [7,9].

Preschool children need healthy teeth to chew and masticate foods, when they graduate from the weaning diet [10,11]. According to the UNICEF multifactorial model of 1990, sufficient dietary intake is among the immediate determinants of children's nutritional status [7].

While some studies conducted among preschool children have documented the link between ECC with the detrimental impacts on the growth and development of a child others have shown weak or no relationship between nutritional status and ECC [11]. Studies conducted by Clarke and colleagues in 2006 [12] and Wasunna in 2012 [13] reported differences in the percentiles of weights of under five years old preschool children with dental caries and without dental caries. Furthermore, studies by Filstrup and colleagues in 2003 [14] documented that pain, discomfort, and irritability attributed to caries have been associated with reduced food intake, disturbed sleeping habits, and impaired secretion of growth hormones. Shreds of evidence on the link between untreated ECC and children's nutritional status remain scanty, controversial, and non-conclusive [8,15].

The reported prevalence of ECC among 3–5 year old's in South Africa [16], Uganda [17] and Kenya [18] ranged between 47% and 63% while in Tanzania the reported prevalence ranged around 30% [11,19]. A slight difference in prevalence of ECC between girls (60%) and boys (52%) aged 3–5 years old was reported in Khartoum- Sudan [20]. Furthermore, the dmft index among 3–5 year old's was reportedly higher in Kenya [18] than in Tanzania [19] and contributed to mainly by the decayed component.

Various factors have been documented in the origin and progression of ECC to preschool children which include; a susceptible host tooth, micro-organisms, diet and time [21]. Among

these factors, diet containing highly fermentable carbohydrates (sweets, ice cream and soda) is a critical factor in the etiology of ECC and preschool children who frequently consume highly fermentable carbohydrates for extended periods have higher risk of developing ECC [22].

Furthermore, studies have shown that environmental factors such as low socio-economic status, lower level of education, the worse children's oral health status and non-exposure to fluoride also play part in the process of ECC [23]. Untreated ECC not only causes pain and discomfort but also places a financial burden to the family and society [24]. In 2013 Mishu and colleagues documented that pain and discomfort experienced among preschool children with untreated ECC affect their chewing abilities leading to practice food selection [15]. Anecdotal evidence suggests that practice of food selection can affect the quantity and quality consumed resulting in under nutrition or over nutrition for those who select easy to eat fast foods [25].

Nutritional assessment includes many factors such as anthropometric measurements of body composition, clinical assessment of altered nutritional requirements and social or psychological issues that may preclude adequate intake [26,27]. Assessment of anthropometric measurements such as stunting, wasting, and underweight to preschool children is prioritized globally [28]. In Tanzania, the prevalence of stunting among preschool children between 2014 and 2018 has decreased significantly from 34.7% to 31.8% [26,27]. However, prevalence of underweight in 2018 was significantly higher as compared to the recorded prevalence in 2014 [26,27].

In the realization of Vision 2025 Global Agenda of well-nourished children for a healthy productive population, assessment of children's nutritional status is required. Furthermore, due to limited studies there is a dearth of information on the association between ECC, risk factors and nutritional status among children in Tanzania. Thus, the present study was carried out with the objective of determining the prevalence of ECC, risk factors and nutritional status among 3-5-year-old preschool children and to assess its correlation with the form, visible plaque scores in upper anterior teeth, total sugar exposure, anthropometric measures, and socio demographic attributes.

Future studies should assess the association between anthropometric measures indicated by weight-for-age and early childhood caries using longitudinal analyses. The findings presented can be utilized in the planning, implementation and evaluation of oral health promotion programs to preschool children.

## Materials and methods

A cross-sectional study was conducted in Kisarawe district, one of the 6 districts in the Pwani Region of Tanzania as shown in Fig 1. The district was conveniently chosen due to its rural (population of 84,174) and semi-urban (population of 17,424) characteristics [29]. The district is administratively divided into 15 wards, namely: Cholesamvula, Kibuta, Kiluvya, Kisarawe, Kurui, Mafizi, Maneromango, Marui, Marumbo, Masaki, Msanga, Msimbu, Mzenga, Vihingo, and Vikumbulu.

The district has approximately 108,398 people, out of whom 3.1% are 3 years old, 3.2% are 4 years old and 3.0% are 5 years old [29] and, had eighty-three registered preschools at the time of the study [30]. Ethical approval for this study was granted by the Senate Research and Publication Committee of the Muhimbili University of Health and Allied Sciences in Tanzania (Ref. No. DA282/298/01.C). Further permission was granted by the District Executive Director and District Education Officer (President's Office, Regional Administrative and Local Government- Tanzania). With assistance of these offices one public preschool from each ward was randomly selected. Respective Head-teachers were informed of the study and were requested to provide each child registered for preschool with a consent form for parents/guardians to

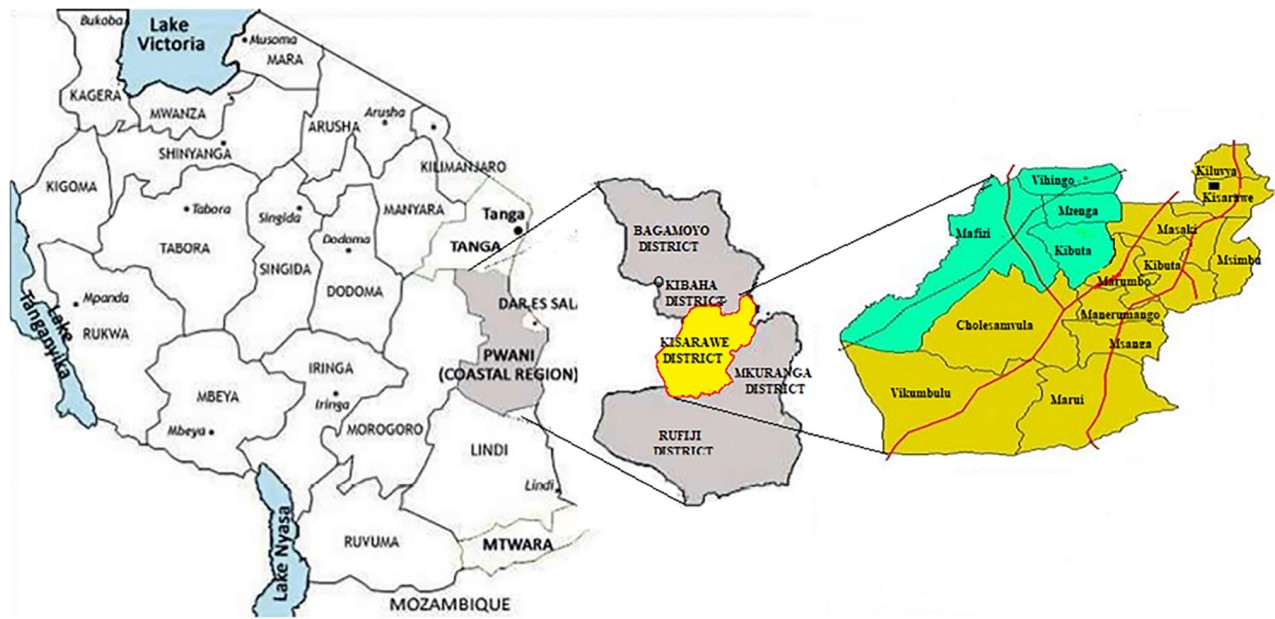

**Fig 1. Map of Tanzania showing Pwani (Coast Region) and Kisarawe District.**

sign and a letter requesting them to escort their child to school on scheduled days. The total sample size of 950 preschool children was sought by assuming the proportion of weight and height, and ECC (outcome) among preschool children in Tanzania of 50%, a margin error of 5%, a confidence level of 95% and a design effect of 2. Another 5% was added to the estimated sample size to account for non-response. Included in the study were a total of 831 preschool children who were present with their parents or guardians and with signed consent to participate in the research.

## Interviews

A standard structured questionnaire was adopted and translated in several steps; from English into Kiswahili by bi-lingual Kiswahili/English professionals, and then translated back to English by independent translators. Kiswahili is the national language spoken proficiently by almost 99% of the Tanzanian population and the reason for its use. Pilot investigation was conducted and the project professionals in the field reviewed the structured questionnaire for semantic, experiential, and conceptual equivalence to the original version. Sensitivity to culture and selection of appropriate words were considered. Trained researchers (TS and LC) interviewed parent or guardian accompanying each child using the structured questionnaire. In summary, the information collected included; socio-demographic factors, dietary habits, and oral health behavior of the children. Parent/guardian level of education and employment was also sought.

Parents' reported children's behaviors on consumption of sugar snacks and drinks (biscuits or cake; candy or chocolate and; sweetened milk or soft drinks) being guided by the following questions: Does your child consume sugary food in between meals (0 = No, 1 = Yes); How many times a day does your child consume sugary food (0 = less than 5 times a day, 1 = more than 5 times a day); In the past 7 days did your child consume sugary foods (0 = never, 1 = one day, 2 = two to four days, 3 = five days and more); What is the amount of sugar teaspoon usually consumed by your child in a day (0 = No, 1 = one to two teaspoon, 2 = more than two

teaspoon); Does your child ever report not being hungry during meal time cause of eating sugary food (0 = No, 1 = Yes); Do you often sweeten your child's milky food with sugar (0 = No, 1 = Yes); Do you often sweeten your child starchy food (e.g. porridge) with sugar (0 = No, 1 = Yes); Does your child consume frequent fruit juices or soda (0 = No, 1 = Yes) and; Does your child consume sweets (e.g. biscuits, cake, candy or chocolate) (0 = No, 1 = Yes). Total sugar consumption was calculated based on the overall intake of frequency. Scores of snacks and drinks were then summed to produce an overall score (range 2–23). The overall score was used to indicate the daily frequency of intake of sugars for each participant. In addition, parents' reported children's daily behavior of consuming fresh fruits and vegetables: Does your child consume fruits (0 = no, 1 = once a day, 2 = twice a day, 3 = thrice a day, 4 = more than 3 times a day) and; Does your child consume vegetables (0 = no, 1 = once a day, 2 = twice a day, 3 = more than 2 times a day). Dummy variables were summarized (range 0–7) and used as a continuous variable. Other oral health behaviors were assessed by asking parents to report on: Do you assist your child to brush (0 = No, 1 = Yes); Do you use fluoride toothpaste while brushing your child (0 = No, 1 = Yes) and; how frequent does your child brush (0 = no, 1 = once a day, 2 = twice a day, 3 = thrice a day, 4 = more than 3 times a day) and then summarized into (0 = No, 1 = Yes). Parent/guardian were requested to provide information on level of education (0 = primary and lower level of education, 1 = secondary and higher level of education) and employment (0 = not formally employed, 1 = formally employed) of both parents/guardians of the child.

## Clinical examination

Calibration exercises for the examiners with respect to presence/absence of visible plaque [31] and early childhood caries were carried out according to the guidelines published by the British Association of the Study of Community Dentistry (BASCD) [32].

Children were clinically examined using a dental mirror and natural light by calibrated dentists (TN and LC) while in a knee-to-knee position. Upper anterior teeth; incisors, lateral incisors and canines were examined visually for presence of plaque which was recorded on the basis of absence of visible plaque = 0 or presence of visible plaque = 1. Dummy variables (0 = No, 1 = Yes) were summarized (range 0–6) and dichotomized into children with a count of a plaque score of 0 as having "good oral hygiene" while children with plaque score of one or more were regarded as having "poor oral hygiene".

Following plaque assessment, visual inspection of children's deciduous teeth for ECC was done after wiping and drying using a piece of sterile gauze. With the aid of a disposable dental mirror, absence/presence of a carious lesion was recorded in the recommended WHO Oral Health Assessment Form for Children, 2013 [33]. A score of 0 denoted absence of any visible carious lesion while a score of 1 denoted presence of visible carious lesion. Missing and filled teeth were given a score of 1. In the present analysis, dmft was used as a count variable following dichotomizing into caries free (dmft = 0) or with caries experience (dmft = 1).

## Nutritional status

Nutritional status among preschool children was assessed by anthropometric measures based on the WHO Growth Standards 2006 [34]. A trained registered nurse from the Muhimbili National Hospital measured and recorded the weight and height of each child. Preschool children were asked to remove their shoes before measurement of weight and height were taken using the Seca weight and height measuring machine (Model 786, Mechanical column scale with large round dial and stadiometer, CE 0123, Vogel and Halke, Germany). Based on the WHO Growth Standards 2006, weight, height and age measures were converted to z-scores,

namely weight-for-age, (WAZ), height-for-age (HAZ) and weight-for-height, (WHZ) [34]. The variables created for underweight (WAZ), stunting (HAZ) and acute malnutrition (WHZ) were used due to its recognition as the best system for analysis and presentation of anthropometric data for preschool children and because of its advantages compared to the other methods [35].

## Reliability of ECC and anthropometric measures

Prior to the fieldwork, two trained dentists (TN and LC) were calibrated for scoring of ECC lesions in preschool children. Calibration was performed using images of different clinical situations on two separate occasions with a one-week interval between sessions. The minimum intra-examiner agreement was 0.83 and the minimum inter-examiner agreement was 0.78. During the fieldwork, duplicate examinations one week apart were performed with 74 preschool children randomly chosen. Intra examiner reliability in terms of Cohen's kappa ranged from 0.81 to 0.89, respectively.

The person who did anthropometric measurements was calibrated prior to the study with an agreement was 0.85. Being blinded to the oral health status, weight and height of participants were measured with a weight **and** height scale to the nearest 0.5 kilograms and 0.5 cm respectively. The equipment was recalibrated daily prior to the procedure. The level of intra-examiner agreement (Cohen's kappa) of 0.72, was considered substantial.

## Statistical analysis

The collected data were entered into the Epi Info version 3.5.4 and ENA Software and further analysis in Predictive Analytics Software, IBM SPSS Statistics version 20. To determine the mean of the children's dmft index and the prevalence of the children with ECC, Pearson Chi-square and One-way ANOVA was used to estimate the relationship between demographic factors (age, sex, and residency) and oral health behaviors (assist the child to brush, child use toothpaste while brushing and, frequency of child tooth brushing). Further, the description of anthropometric measurement (actual weight and height, and z-score for weight-for-age, height-for-age and weight-for-height), visible plaque score index, sugar consumption index, fresh fruit consumption index, and ECC (dmft index) among preschool children was determined.

To control the effect of possible confounding factors, all variables were entered into multiple logistic regression and Poisson regression models with the dmft >1 and, dmft index as dependent variables. Moreover, a simple linear regression model with the children's dmft index as the dependent variable was constructed to explore the relationship with weight (in Kg), height (in cm), age (in months), weight-for-age z-score (WAZ), height-for-age z-score (HAZ), weight-for-height z-score (WHZ), Visible plaque score index, sugar consumption and, fresh fruit consumption. An alpha level of 0.05 was considered as statistical significance. The multilevel linear analysis was further constructed to explore the association between dmft index (dependent variable) and, weight-for-age z-score, height-for-age z-score, weight-for-height z-score (independent variables), adjusted with all variables associated with either dmft >1 or dmft index (visible plaque score index, sugar consumption, residence, mothers working status, assist child to brush and, child use toothpaste during brush). The same sets of models were used to assess the bidirectional association between dmft index (independent variable) and weight-for-age z-score, height-for-age z-score, weight-for-height z-score (dependent variables).

## Results

A total of 831 participants were recruited in this study from the 950 invited to participate, giving a response rate of 87.5%. Although 80.4% (n = 668) and 86.3% (n = 717) of preschool fathers and mothers had primary education, more than ninety percent of fathers and mothers (n = 746 and n = 795) were not formally employed. The mean age of the participants was 50.34 (±8.41) months with almost equal male/female ratio [male = 48.3% (n = 401) and female = 51.7 (n = 430)]. Mother parents 67.9% (n = 564) reported not to assist their children tooth brushing and 81.0% (n = 673) reported their children to use toothpaste while brushing. However only 23.1% (n = 192) reported their children to brush more than once in a day.

### Children's oral health status

According to the clinical oral examination, prevalence of ECC was 44.8%. Children with one to four, and more than five decayed teeth were 47.7% (n = 219), and 21.8% (n = 101), respectively. The dmft of 2.51 was mostly contributed to by the decayed component. More than half [56.1% (n = 466)] of the participating preschool children had visible plaque on buccal surfaces of upper anterior teeth. Of the 56.1% children with visible plaque, 11.8% (n = 55), 37.8% (n = 176), and 50.4% (n = 235) had one, two, and more than two teeth with visible plaque, respectively.

### Results of anthropometric measures

Participants in this study had a prevalence of underweight ($<$-2 z-score) of 4.2% [95% CI (3.0–5.8)], stunting ($<$-2 z-score) of 1.6% [95% CI (0.9–2.7)] and global acute malnutrition ($<$-2 z-score and/or oedema) of 29.8% [95% CI (26.8–33.0)] as shown in Table 1. Also observed was severe underweight ($<$-3 z-score) of 0.2% [95% CI (0.1–0.9)], severe stunting ($<$-3 z-score) of 0.4% [95% CI (0.1–1.1)] and severe acute malnutrition ($<$-3 z-score and/or oedema) of 5.1% [95% CI (3.8–6.8)].

The distribution of Weight-for-Age of preschool children aged 3 to 5 years in Kisarawe as shown in Fig 2 is shifted to the left but still following the WHO standard natural distribution of reference population when WHO flags are applied with mean z-score -0.39 ± 1.06 Standard deviation, SD.

The Fig 3 displays a distribution of preschool children in Kisarawe in terms of Height-for-Age that is shifted to the right and flatter in comparison to the WHO standard normal distribution of reference population even when WHO flags are applied. The mean HAZ was 0.57 ± 1.08. The flattened distribution may have been contributed to by difficulties encountered during data collection especially on age estimation.

Shown in Fig 4 is the distribution of Weight-for-Height that is shifted to the left but still following the WHO standard normal distribution of reference population, with mean WHZ -1.18±1.45 SD.

### Univariate analysis of risk factors associated with dmft $\geq$1 and dmft score index

In univariate analysis as presented in Table 2, chi-square test showed that children whose residency are semi-rural and having visible plaque on upper anterior teeth were both significantly associated (p$<$ 0.001) with the prevalence of ECC (dmft $\geq$1).

Further, the one-way between groups analysis of variance (ANOVA) conducted to explore the impact of various risk factors on the mean score of dmft index showed a statistically significant difference at the p $<$ 0.001 level in dmft index scores at two residency groups F (1, 616) =

**Table 1. The anthropometric measures, visible plaque score index, sugar consumption index and fresh fruit consumption index distributions among preschool children aged 3 to 5 years in Kisarawe.**

| No | Variable | Mean ± Standard deviation, SD | Prevalence (95% Confidence interval, CI) |
|----|----------|-------------------------------|------------------------------------------|
| 1 | Weight (in Kg) | 15.82 ± 2.10 | |
| 2 | Height (in cm) | 106.6 ± 6.67 | |
| 3 | Age (in months) | 50.34 ± 8.407 | |
| 4 | Weight-for-age z-score (WAZ) | | 4.2% (3.0–5.8) |
| | • Prevalence of underweight ($<$-2 z-score) | | 4.0% (2.8–5.5) |
| | • Prevalence of moderate underweight ($<$-2 z-score and $> =$ -3 z-score) Prevalence of severe underweight ($<$-3 z-score) | | 0.2% (0.1–0.9) |
| 5 | Height-for-age z-score (HAZ) | | 1.6% (0.9–2.7) |
| | • Prevalence of stunting ($<$-2 z-score) | | 1.2% (0.7–2.2) |
| | • Prevalence of moderate stunting ($<$-2 z-score and $> =$ -3 z-score) Prevalence of severe stunting ($<$-3 z-score) | | 0.4% (0.1–1.1) |
| 6 | Weight-for-height z-score (WHZ) | | 29.8% (26.8–33.0) |
| | • Prevalence of global acute malnutrition ($<$-2 z-score and/or oedema) | | 24.7% (21.9–27.8) |
| | • Prevalence of moderate acute malnutrition ($<$-2 z-score and $> =$ -3 z-score, no oedema) Prevalence of severe acute malnutrition ($<$-3 z-score and/or oedema) | | 5.1% (3.8–6.8) |
| 7 | Visible plaque score index | 1.40 ± 1.38 | |
| 8 | Sugar consumption | 15.45 ± 2.12 | |
| 9 | Fresh fruit consumption | 2.83 ± 1.08 | |

6.37, p $<$ 0.001. Despite reaching statistical significance, the actual difference in mean scores between groups was quite small. The effect size, calculated using eta squared, was 0.01.

A simple linear regression was carried out to all variables in Table 1, namely, weight (in Kg), height (in cm), age (in months), weight-for-age z-score, height-for-age z-score, weight-for-height z-score, visible plaque score index, sugar consumption scores and, fresh fruit consumption scores to predict dmft score index. The results of the regression (not shown in the table) indicated that the dmft score index was negatively and significantly associated with weight-for-age z-score (Coefficient: -0.07; 95% CI: -0.61, 0.00) in the unadjusted model. Moreover, there were significant increases in visible plaque scores index and sugar consumption

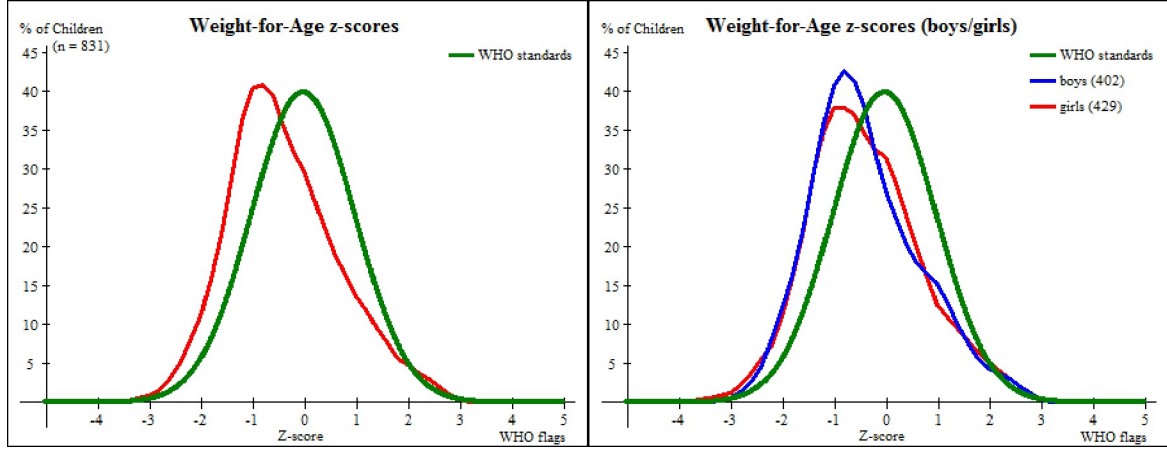

**Fig 2. A comparison of weight-for-age z-score distributions between preschool children in Kisarawe and the WHO standards.**

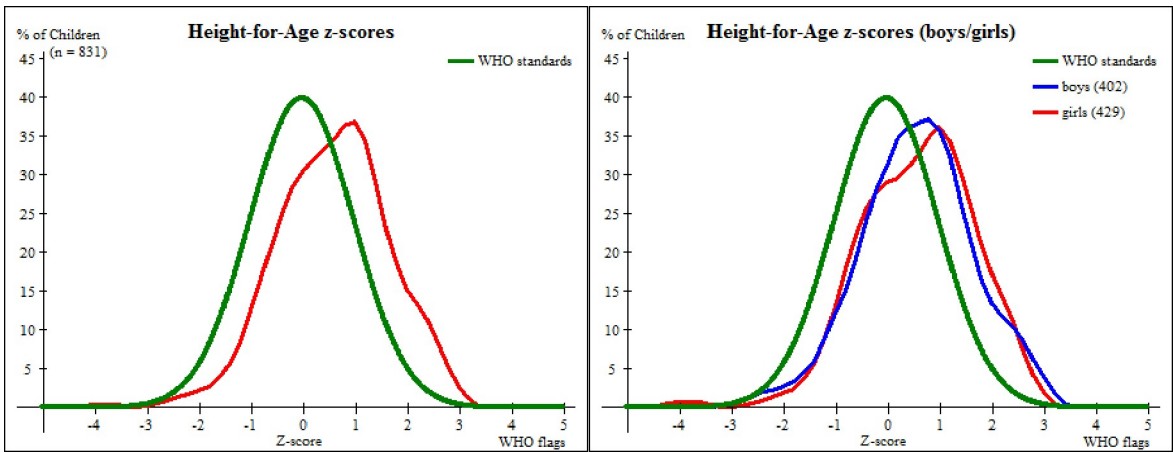

**Fig 3. A comparison of height-for-age z-score distributions between preschool children in Kisarawe and the WHO standards.**

scores with regression coefficient of 0.18 (95% CI: 0.22, 0.56) and 0.17 (95% CI: 0.13, 0.36), respectively.

## Multivariate logistic regression, Poisson regression and linear regression analysis of risk factors associated with dmft ≥1 and dmft score index

As shown in Table 2, lower prevalence of ECC (dmft ≥1) had been associated with children whose residency where from semi-rural [adjusted OR 0.55 (95% CI: 0.39, 0.79)] and, higher prevalence from children with presence of visible dental plaque [adjusted OR 2.97 (95% CI: 2.23, 3.96)]. Poisson regression was employed to rule out the risk of losing precision when dmft ≥1 employed in the multiple variable logistic regression analysis. The dmft scores index shows significant association with residence [adjusted RR 1.34 (95% CI: 1.15, 1.56)], mothers working condition [adjusted RR 1.65 (95% CI: 1.19, 2.27)], do you assist your child to brush? [Adjusted RR 0.85 (95% CI: 0.77, 0.95)], does your child use toothpaste while brushing? [Adjusted RR 1.18 (95% CI: 1.04, 1.34)] and, presence of visible plaques [adjusted 0.69 (95% CI: 0.58, 0.73)].

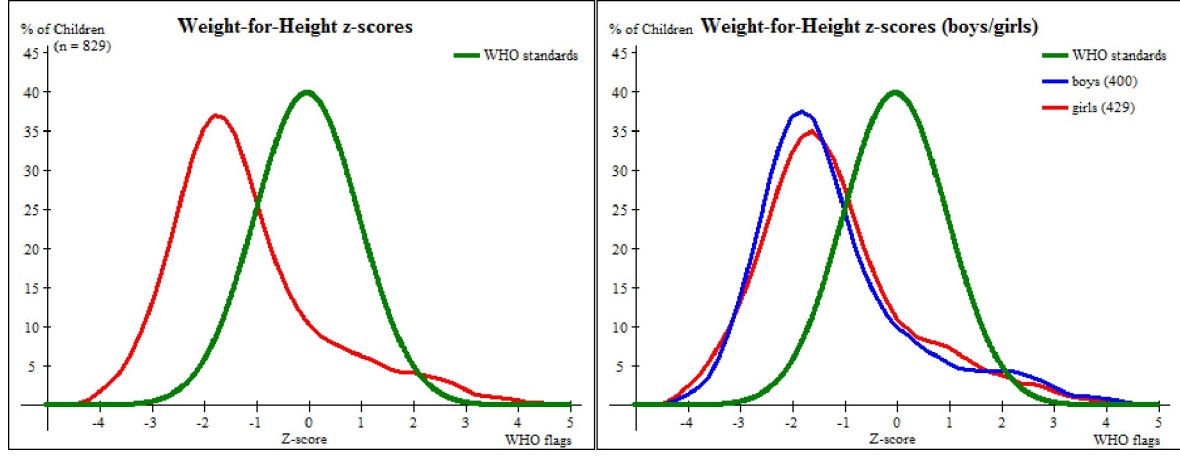

**Fig 4. A comparison of weight-for-height z-score distributions between preschool children in Kisarawe and the WHO standards.**

**Table 2. Univariate and multivariate analysis of different factors associated with dental caries status in deciduous tooth (n = 831).**

| Variables | Categories | Univariate analysis | | Multivariate analysis | |
| | | dmft ≥1 | dmft index | dmft ≥1 | dmft index |
| | | [1a] Caries prevalence % (n) | [1b] F- test (df) | [1c] OR, 95% CI | [1d] RR, 95% CI |
| Child sex | Male | 50.3 (187) | 0.85 (1) | 1 | 1.08 (0.98, 1.19) |
| | Female | 49.7 (185) | | 1.11 (0.85, 1.46) | 1 |
| Child age | 3 years | 3.2 (12) | 0.41 (2) | 1 | 1.12 (0.88, 1.42) |
| | 4 years | 18.8 (70) | | 1.48 (0.65, 3.37) | 0.92 (0.81, 1.05) |
| | 5 years | 78.0 (290) | | 1.02 (0.47, 2.18) | 1 |
| Residence | Rural | 84.7 (315) | 6.37 (1) *** | 1 | 1.34 (1.15, 1.56) *** |
| | Semi-rural | 15.3 (57) *** | | 0.55 (0.39, 0.79) *** | 1 |
| Mothers level of education | Informal and Primary education | 86.6 (322) | 2.48 (1) | 1 | 1.034 (0.8, 1.25) |
| | Secondary education and above | 13.4 (50) | | 0.96 (0.64, 1.43) | 1 |
| Fathers level of education | Informal and Primary education | 79.8 (295) | 1.81 (1) | 1 | 1.11 (0.95, 1.29) |
| | Secondary education and above | 20.2 (75) | | 0.72 (1.07, 1.58) | 1 |
| Mothers working condition | Unemployed | 96.2 (358) | 4.03 (1) * | 1 | 1.65 (1.19, 2.27) *** |
| | Employed | 3.8 (14) | | 0.78 (0.39, 1.53) | 1 |
| Fathers working condition | Unemployed | 90.1 (335) | 0.92 (1) | 1 | 0.95 (0.78, 1.15) |
| | Employed | 9.9 (37) | | 0.95 (0.60, 1.49) | 1 |
| Do you assist your child to brush? | No | 64.5 (240) | 0.50 (1) | 1 | 0.85 (0.77, 0.95) *** |
| | Yes | 35.5 (132) | | 1.32 (0.98, 1.77) | 1 |
| Does your child use toothpaste while brushing? | No | 18.3 (68) | 3.23 (1) | 1 | 1.18 (1.04, 1.34) *** |
| | Yes | 81.7 (304) | | 1.09 (0.77, 1.55) | 1 |
| Frequent child tooth brushing | Not every day | 3.0 (11) | 0.10 (2) | 1 | 1.10 (0.79, 1.54) |
| | Once a day | 73.9 (275) | | 1.04 (0.45, 2.45) | 1.11 (0.98, 1.25) |
| | Twice or more a day | 23.1 (86) | | 1.01 (0.73, 1.39) | 1 |
| Visible plaques dichotomous score | No | 29.6 (110) *** | 16.08 (1) | 1 | 0.69 (0.58, 0.73) *** |
| | Yes | 70.4 (262) | | 2.97 (2.23, 3.96) *** | 1 |

[1a] Pearson Chi-square test.

[1b] One Way Anova.

[1c] Logistic regression.

[1d] Poisson regression.

*** p<0.01.

* p<0.05.

As shown in Table 3, a multilevel linear analysis to explore the association between change dmft score index and the anthropometric measures adjusted with visible plaque score index, sugar consumption, residence, mothers working status, assist child to brush and, child use toothpaste during brush revealed that: the dmft score index was negatively and significantly associated with weight-for-age z-score [Coefficient: -0.11 (95% CI: -0.76, -0.11)] and, positively significantly associated with visible plaque score index [Coefficient: 0.16 (95% CI: 0.18, 0.52)]

**Table 3. Multilevel linear analysis of factors associated with dmft score index.**

|  | Dmft score index |
| --- | --- |
| **Variable** | **Coefficient (95% CI)** |
| Weight for age z-score | -0.11 (-0.76, -0.11) *** |
| Height for age z-score | -0.07 (-0.59, 0.05) |
| Weight for height z-score | -0.05(-0.41, 0.08) |
| Visible plaque index | 0.16 (0.18, 0.52) *** |
| Sugar consumption | 0.18 (0.14, 0.37) *** |
| Residence | -0.07 (-1.16, 0.09) |
| Mothers working status | -0.07 (-2.03, 0.09) |
| Do you assist your child to brush | 0.07 (-0.08, 0.95) |
| Does your child use toothpaste during brush | -0.07 (-1.19, 0.09) |

*** $p < 0.01$.

* $p < 0.05$.

and sugar consumption score [Coefficient: 0.19 (95% CI: 0.15, 0.38)] in the adjusted model. However, dmft score index was not statistically significant with height-for age z-score and weight for height z-score.

In Table 4 we further explored the bilateral relation of weight-for-age z-score, height-for age z-score and, weight for height z-score (as dependent variables) and dmft score index (as dependent variable) adjusted with visible plaque score index, sugar consumption, residence, mothers working status, assist child to brush and, child use toothpaste during brushing revealed that: weight-for-age z-score were still negatively and significantly associated with dmft score index [Coefficient: -0.11(95% CI: -0.05, -0.01)]. However, height-for age z-score and, weight for height z-score were not statistically significant with dmft score index.

## Discussion

Early childhood caries (ECC) is a significant pediatric health problem in both developed and developing countries [1]. This population based cross-sectional study was conducted in Kisarawe, Tanzania with the objective of determining the prevalence of ECC, risk factors and nutritional status among 3-5-year-old preschool children and to assess its correlation with the form,

**Table 4. Multilevel linear analysis of factors associated with changes in weight for age z-score, height for age z-score and weight for height z-score.**

|  | Weight for age z-score | Height for age z-score | Weight for height z-score |
| --- | --- | --- | --- |
| **Variable** | **Coefficient (95% CI)** | **Coefficient (95% CI)** | **Coefficient (95% CI)** |
| dmft index | -0.11(-0.05, -0.01) *** | -0.07 (-0.04, 0.003) | -0.06 (-0.04, 0.01) |
| Visible plaque index | 0.06 (-0.01, 0.07) | 0.04 (-0.02, 0.06) | 0.05 (-0.02, 0.09) |
| Sugar consumption | 0.09 (0.01, 0.06) * | 0.09 (0.003, 0.06) * | 0.05 (-0.02, 0.06) |
| Residence | 0.08 (0.01, 0.32) * | 0.07 (-0.02, 0.30) | 0.05 (-0.07, 0.34) |
| Mothers working status | 0.01 (-0.23, 0.29) | 0.03 (-0.16, 0.37) | 0.02 (-0.25, 0.44) |
| Do you assist your child to brush | -0.07 (-0.24, 0.02) | -0.09 (-0.27, -0.02) * | -0.07 (-031, 0.02) |
| Does your child use toothpaste during brush | -0.08 (-0.30, 0.02) | -0.06 (-0.28, 0.04) | 0.04 (-0.11, 0.31) |

*** $p < 0.01$.

* $p < 0.05$.

visible plaque scores in upper anterior teeth, total sugar exposure, anthropometric measures, and socio demographic attributes. ECC affects 60–90% of preschool children throughout the world [36] and over 90% of ECC remains untreated accompanied with discomfort or tooth-ache affecting young children's growth and well-being [10]. There is dearth of information in Sub- Saharan African countries on how untreated carious lesions does affect the nutritional status and anthropometric measures of preschool children [37]. The African Regional strategy on Oral health 2016–2025 emphasized on shifting the focus of dental research from the causes of the dental diseases to how dental diseases affect general health [38].

The present study revealed a considerable higher prevalence of ECC (44.8%) among children of 3 to 5 years old than that documented in the study of Rwakatema and Ng'ang'a in Moshi, Tanzania [19] and Nobile and colleagues in Southern Italy [39]. The mean dmft index score in this study was slightly similar to that documented by Musinguzi and colleagues in the rural community of Uganda [24]. Generally, the reported prevalence in this study is substantially higher than the prevalence reported in most developed countries [40]. The low prevalence of ECC in most developed countries is much contributed by the implementation of effective integrated preventive policies and programs, which should be considered in developing countries [41].

Various authors have documented the association between parent's level of education and ECC [2], however, this study could not establish any association between parent's levels of education and ECC. In contrast, other studies reported an association between ECC and a lower level of parents education [42]. Comparison of these studies should be done with caution because of differences in study methodology, such as the study design and the methods for assessing ECC [43].

Moreover, in the present study, preschool children who resided in rural areas experienced greater impact of ECC in comparison to those residing in semi-rural areas. Similar findings were documented by other studies [11,44]. Few studies conducted in the sub Saharan African countries could not establish such an association [45,46]. Inadequate oral health care information and inefficient oral hygiene practices seem to explain the high levels of ECC among preschool children residing in rural areas.

Good oral hygiene is one of the prerequisites for preventing ECC to preschool children [47]. However, the majority of the preschool children in the present study unveiled poor oral hygiene, and only 32.1% of parents reported supervising their preschool children during tooth brushing. Thus, the strongest positive correlation between ECC and visible plaque [Coefficient: 0.16 (95% CI: 0.18–0.52)] in this study may be explained by the lack of accompaniment by parents during tooth brushing or unawareness to oral hygiene practices. Similar findings were documented in another sub Saharan African country [48].

In the present study, the total number of sugar exposure was found to be directly related to ECC [Coefficient: 0.18 (95% CI: 0.14, 0.37)] among preschool children aged 3 to 5 years old. These results suggest that there is direct correlation of total sugar intake with the prevalence of ECC. Similar findings were also reported by Sreebny [49], however, Yabao and colleagues [50] reported that there was no significant relationship between sugar intake and dental caries. Undoubtedly, there is a strong correlation between these two variables with an increase in one factor leading to an increase in the other. There is an urgent need for the dissemination of appropriate and accurate information on healthy nutrition and oral health care for preschool children.

This study demonstrated a direct negative correlation between ECC and weight-for-age z-score (WAZ), [Coefficient: -0.11 (95% CI: -0.76, -0.11)]. Some studies have documented such kind of association [51,52], while other studies reported a lack of association [53]. Studies

argued that WAZ among preschool children would result in ECC increment [51], which was in accordance with the results of this study.

The global information on the association between WHZ and ECC is inconclusive and controversial. Some studies have shown significant associations while others show weak or no relationship [53]. In this study, we could not establish an association between WHZ and preschool ECC. On the contrary, a study conducted in Sub-Saharan Africa reported an association between WHZ and ECC and closely linked it with poverty as a main contributory factor [51]. Further in this study we could not establish any relationship between HAZ and ECC.

The observed relationship between ECC and WAZ and HAZ could also be confounded by other factors. It is worth noting that the bidirectional associations between ECC on one hand, and WAZ and HAZ on the other were independent from a number of risk factors related to the development of dental caries such as sugar consumption, oral hygiene and family residence [2].

The findings of this study have some implications to the health policies of developing countries experiencing the double burden of ECC and poor nutritional status among preschool children such as in Tanzania. Health policies and programs in Tanzania prioritize life threatening diseases in comparison to what is considered less life threatening such as ECC [54]. Undoubtedly, the potential impact of ECC on WAZ observed in this study and other studies in developing countries points out the need for prioritizing integrated programs for oral health care and nutritional aspects of preschool children, not only for pain relieve but also for preventing the negative impact on children's growth and general wellbeing [55].

Several limitations could be found in the present study. Firstly, the major limitation of this study relates to the study design and the fact that these results cannot be generalized to the entire population. This is because the sample calculation was drawn from preschool children enrolled in public schools however it provides baseline data for further studies. Secondly, some potential factors that contributed to ECC, such as poverty and/or socio-economic status and access to dental care were not included in this study. We did not survey the socio-economic status because during the pilot investigation many parents were observed not to be in formal employment and expressed uncertainty in determining value of fixed assets owned. Thirdly, there may have been an unavoidable recall bias from the children's parents in responding to questions. Finally, the present study was a cross-sectional investigation, so causal inference was limited.

## Conclusions

This study demonstrated a significantly negative relationship between ECC and children's anthropometric measures indicated by weight-for-age, and positive relationship with sugar exposure and poor oral hygiene indicated by visible plaque on upper anterior teeth. Controlling risk factors will reduce the occurrence of ECC thereby catering for a healthy population of well-nourished children, in line with the realization of vision 2025 global agenda.

It is recommended that future studies should assess the association between anthropometric measures indicated by weight-for-age and early childhood caries using longitudinal studies. The findings presented can be utilized in the planning, implementation and evaluation of oral health promotion programs to preschool children.

## Supporting information

**S1 File.**
(SAV)

## Acknowledgments

We acknowledge the Muhimbili University of Health and Allied Sciences, the administrative authorities in Kisarawe District, preschool children for their participation, teachers, parents and all of those with whom we had the pleasure to work during this project.

## Author Contributions

**Conceptualization:** Lorna C. Carneiro, Ray M. Masumo.

**Data curation:** Tumaini S. Ndekero, Lorna C. Carneiro, Ray M. Masumo.

**Formal analysis:** Ray M. Masumo.

**Funding acquisition:** Tumaini S. Ndekero.

**Methodology:** Lorna C. Carneiro, Ray M. Masumo.

**Project administration:** Tumaini S. Ndekero.

**Resources:** Tumaini S. Ndekero.

**Software:** Ray M. Masumo.

**Supervision:** Lorna C. Carneiro.

**Validation:** Lorna C. Carneiro, Ray M. Masumo.

**Writing – original draft:** Tumaini S. Ndekero.

**Writing – review & editing:** Lorna C. Carneiro, Ray M. Masumo.

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
