## [Decision Letter · Decision Letter 0]

8 Dec 2020

PONE-D-20-31062

Prevalence of early childhood caries, risk factors and nutritional status among 3-5-year-old preschool children in Kisarawe, Tanzania

PLOS ONE

Dear Dr. CARNEIRO,

Thank you for submitting your manuscript to PLOS ONE. After careful consideration, we feel that it has merit but does not fully meet PLOS ONE’s publication criteria as it currently stands. Therefore, we invite you to submit a revised version of the manuscript that addresses the points raised during the review process.

We look forward to receiving your revised manuscript.

Kind regards,

Ove A. Peters, DMD MS PhD

Academic Editor

PLOS ONE

Journal Requirements:

2.In your Data Availability statement, you have not specified where the minimal data set underlying the results described in your manuscript can be found. PLOS defines a study's minimal data set as the underlying data used to reach the conclusions drawn in the manuscript and any additional data required to replicate the reported study findings in their entirety. All PLOS journals require that the minimal data set be made fully available. For more information about our data policy, please see http://journals.plos.org/plosone/s/data-availability.

3.Thank you for stating the following financial disclosure:

 [The funders had no role in study design, data collection and analysis, decision to publish, or preparation of the manuscript.].

4. We noticed you have some minor occurrence of overlapping text with the following previous publications, which needs to be addressed:

- https://kclpure.kcl.ac.uk/ws/files/109319328/The_bidirectional_relationship_between_SHEN_Accepted16April2019_GOLD_VoR_CC_BY_.pdf

- http://www.scielo.br/scielo.php?script=sci_arttext&pid=S1806-83242013000400356&lng=en&nrm=iso&tlng=en

In your revision ensure you cite all your sources (including your own works), and quote or rephrase any duplicated text outside the methods section. Further consideration is dependent on these concerns being addressed.

Additional Editor Comments (if provided):

This ms has been reviewed by two experts and while it does have merits it requires major changes. Please see the attached reports and address all comments in a potential resubmission.

Reviewers' comments:

Reviewer's Responses to Questions

**Comments to the Author**

1. Is the manuscript technically sound, and do the data support the conclusions?

Reviewer #1: No

Reviewer #2: Partly

2. Has the statistical analysis been performed appropriately and rigorously? 

Reviewer #1: No

Reviewer #2: I Don't Know

3. Have the authors made all data underlying the findings in their manuscript fully available?

Reviewer #1: Yes

Reviewer #2: Yes

4. Is the manuscript presented in an intelligible fashion and written in standard English?

Reviewer #1: No

Reviewer #2: Yes

5. Review Comments to the Author

Reviewer #1: I have annotated the submitted manuscript. I unfortunately struggled with understanding the aim of the research, the complex analysis and presentation of the findings. The discussion did have a clear message for oral health in Tanzania; and lacked any feasible recommendations.

Reviewer #2: This is an interesting study on the prevalence of ECC and the risk factors involved and is presented in a mostly intelligible manner, however some parts of the manucsript could be made clearer and more concise.

See attachment for specific comments.

6. PLOS authors have the option to publish the peer review history of their article (what does this mean?). If published, this will include your full peer review and any attached files.

Reviewer #1: No

Reviewer #2: No

---

## [Author Response · Author response to Decision Letter 0]

2 Feb 2021

The authors extend their appreciation to the reviewers and acknowledge that all comments have been addressed and uploaded as 'response to reviewers'

---

## [Editor Report · Decision Letter 1]

4 Feb 2021

Prevalence of early childhood caries, risk factors and nutritional status among 3-5-year-old preschool children in Kisarawe, Tanzania

PONE-D-20-31062R1

Dear Dr. CARNEIRO,

We’re pleased to inform you that your manuscript has been judged scientifically suitable for publication and will be formally accepted for publication once it meets all outstanding technical requirements.

Kind regards,

Ove A. Peters, DMD MS PhD

Academic Editor

PLOS ONE

Additional Editor Comments (optional):

Thank you for the careful revision.
---

## [Editor Report · Acceptance letter]

8 Feb 2021

PONE-D-20-31062R1 

Prevalence of early childhood caries, risk factors and nutritional status among 3-5-year-old preschool children in Kisarawe, Tanzania. 

Dear Dr. Carneiro:

I'm pleased to inform you that your manuscript has been deemed suitable for publication in PLOS ONE. Congratulations! Your manuscript is now with our production department. 

Kind regards, 

on behalf of

Dr. Ove A. Peters 

Academic Editor

PLOS ONE